# Computational Analysis of PDE-Based Shape Analysis Models by Exploring the Damped Wave Equation

Alexander Köhler *,† and Michael Breuß †

Institute of Mathematics, Brandenburg University of Technology (BTU) Cottbus-Senftenberg, Platz der Deutschen Einheit 1, 03046 Cottbus, Germany
* Correspondence: koehlale@b-tu.de
† These authors contributed equally to this work.

**Abstract:** The computation of correspondences between shapes is a principal task in shape analysis. In this work, we consider correspondences constructed by a numerical solution of partial differential equations (PDEs). The underlying model of interest is thereby the classic wave equation, since this may give the most accurate shape matching. As has been observed in previous works, numerical time discretisation has a substantial influence on matching quality. Therefore, it is of interest to understand the underlying mechanisms and to investigate at the same time if there is an analytical model that could best describe the most suitable method for shape matching. To this end, we study here the damped wave equation, which mainly serves as a tool to understand and model properties of time discretisation. At the hand of a detailed study of possible parameters, we illustrate that the method that gives the most reasonable feature descriptors benefits from a damping mechanism which can be introduced numerically or within the PDE. This sheds light on some basic mechanisms of underlying computational and analytic models, as one may conjecture by our investigation that an ideal model could be composed of a transport mechanism and a diffusive component that helps to counter grid effects.

**Keywords:** shape analysis; partial differential equations; damped wave equation; wave equation; time discretisation; numerical methods; Laplace–Beltrami operator





## 1. Introduction

The computation of shape correspondences is a fundamental task in computer vision with many potential applications [1,2]. In the setting of three-dimensional shape analysis, the underlying problem amounts to identifying an explicit relation between the surface elements of two or more shapes. The variety of possible shape correspondence mappings that is of interest in applications includes non-rigid transformations where shapes are just almost isometric, allowing, e.g., to match different poses of human or animal shapes.

An important solution strategy is to achieve a pointwise shape correspondence using so-called descriptor-based methods. For this, a feature descriptor has to be computed that characterises each point on a shape by describing the surrounding shape surface geometry. A mathematically sound approach to compute such shape signatures is to make use of the spectral decomposition of the Laplace–Beltrami operator; see, e.g., [2–4]. To this end, the Laplace–Beltrami operator may be incorporated in a certain variety of partial differential equations (PDEs) that are potentially useful as models for feature computation. The arguably most important classic signature that can be obtained in this way is the heat kernel signature (HKS) [4], which relies on the heat equation; however, this also includes versions of the Schrödinger equation leading to the wave kernel signature (WKS) [5]. Additionally, the hyperbolic wave equation [6] has already been proposed.

For the computation of such PDE-based signatures, the basic task amounts to resolving the underlying PDEs on a manifold representing a shape's boundary. The HKS and WKS

are both kernel-based methods that rely on the eigenfunction expansion of the Laplace–Beltrami operator to tackle this task and to achieve efficient algorithms. An alternative to the spectral approach is to consider the numerical integration of the underlying PDEs as proposed in [6–8]. As has been shown in the mentioned works, the shape-matching accuracy of the numerical descriptors constructed in the latter works may be slightly higher than by the kernel-based approaches, such as HKS. However, when following the path towards numerical feature construction, it is highly advocated to employ very efficient computational means such as the model order reduction framework presented in [8,9] in order to avoid high computational times.

In previous works based on numerical integration, the first-order implicit time integration has been studied in detail [7]. For the wave equation model, it has been shown that backward differencing in time may yield favourable results over simple central differences [6]. In particular, as has been illustrated in [6], the classic wave equation may yield in some experiments results of higher corresponding quality compared to the other mentioned models. Let us note that this holds again in particular when assessing the models using numerical integration with implicit first-order time stepping.

**Our Contributions**   In this paper, we give an account of efforts on analysing important numerical aspects of PDE-based shape descriptors. According to the previous results described above, the question arises if some implicit schemes offer particularly useful key properties in the shape analysis context. To answer this question, we present a detailed study of three implicit time stepping schemes for the wave and damped wave equation on manifolds. We show that standard finite differences may not be appropriate since the typical initial condition used for the construction of shape signatures, which is a discrete Dirac delta function, may yield oscillatory artefacts that may spoil correspondence quality. In order to resolve this issue, we show that $l_0$-stable methods should be employed, extending the first results presented in [10] in several ways by a much more detailed exposition. We explore in detail that, in addition, a certain amount of numerical or explicit diffusion is beneficial to achieve results of best matching quality. To perform the study, we have developed a unified numerical model order reduction framework based on [8]. The experimental investigation is performed at the hand of dedicated synthetic settings and selected shape data sets in order to show in detail the important effects.

As for previous work relating to this paper, the closest publication we would like to mention is surely our conference paper [10], where the $l_0$-stability of schemes also used in the current work is studied in detail for heat and wave equations. We actually employ some information on those results in some part of this paper, as visible in some remarks with clear citation. Furthermore and naturally, we recall the numerical schemes in a similar style here. In all other aspects, we begin our investigations here at the point where [10] concludes.

## 2. Theoretical Background

Talking about shape correspondence and shape matching, we need to build a formal understanding of shapes and an idea of comparing them. To this end, imagine, e.g., two versions of a centaur and one version of a dog. It appears obvious that the two different versions of a centaur, such as standing and bowing as in Figure 1, have more in common than a centaur and the dog. Naturally, we expect to also obtain corresponding results from our numerical model. Therefore, we study the behaviour of wave propagation on shapes. This propagation will lead to an object called *feature descriptor*. The descriptors can be compared pointwise with each other to find similarities between the shapes. In the end, we should have a better match between two centaur shapes than between a dog and a centaur shape.

This idea will be realised by mathematical formulas over the next sections. The geometric basics of the set-up coincide with the classical framework in the field; see, e.g., [2].

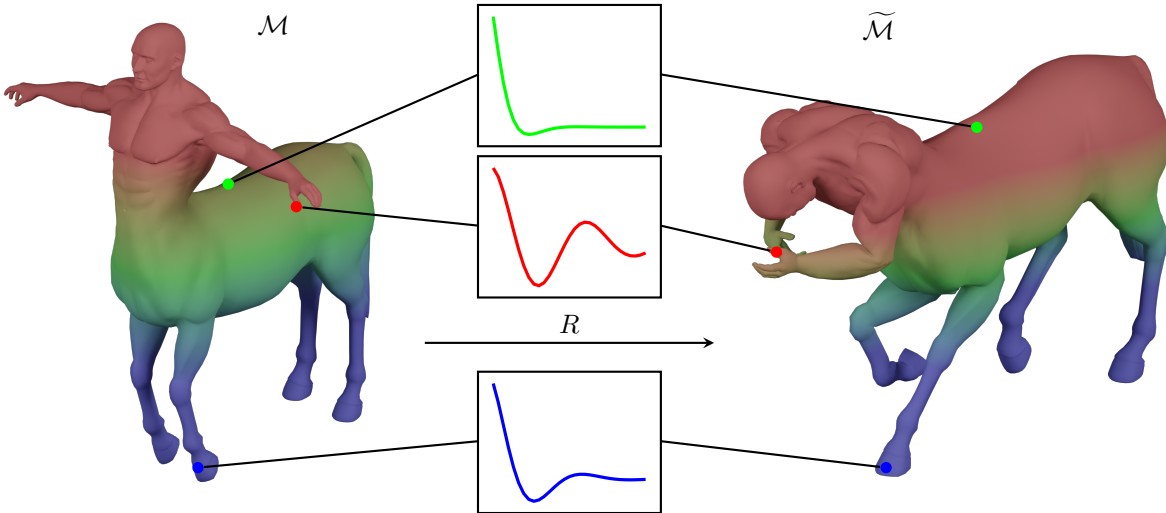

**Figure 1.** Almost isometric transformation $T$ between two centaur shapes $\mathcal{M}$ (shape 0) and $\widetilde{\mathcal{M}}$ (shape 2) of the TOSCA dataset [2]. Additionally, we see the feature descriptor (function plots in the boxes) at three different points $x_i$ with $i \in \{13, 221, 2686, 3269\}$. These are located at the left thumb, the back and the left front hoof, in this order. The plotted feature descriptors are created for illustration of the underlying idea and are not based on real calculations.

### 2.1. Basics on Geometry of Shapes

We consider a shape $\mathcal{M} \subset \mathbb{R}^3$ as a compact two-dimensional Riemannian manifold of a three-dimensional geometric object $\mathcal{B}$, and we assume that it is declared by its bounding surface $\mathcal{M} = \partial \mathcal{B}$. Two shapes $\mathcal{M}$ and $\widetilde{\mathcal{M}}$ will be called isometric if there is a smooth homeomorphism $S \colon \mathcal{M} \to \widetilde{\mathcal{M}}$ between both shapes that preserves the intrinsic distance $d_{\mathcal{M}} \colon \mathcal{M} \times \mathcal{M} \to \mathbb{R}$

$$d_{\mathcal{M}}(x_i, x_j) = d_{\widetilde{\mathcal{M}}}(S(x_i), S(x_j)) \tag{1}$$

for all surface points $x_i, x_j \in \mathcal{M}$.

Considering only isometric shapes would be far too restrictive in our framework. Nearly all shapes have small elastic deformations when they undergo a transformation that will undermine the equality in Equation (1). These deformations occur either as a by-product of shape acquisition or due to the variability in the pose itself; see Figure 1. With this in mind, we use the almost isometric transformation $R \colon \mathcal{M} \to \widetilde{\mathcal{M}}$, which will allow small deformations

$$\left| d_{\mathcal{M}}(x_i, x_j) - d_{\widetilde{\mathcal{M}}}(R(x_i), R(x_j)) \right| < \varepsilon \tag{2}$$

for all $x_i, x_j \in \mathcal{M}$ and $\varepsilon$ a small non-negative number.

### 2.2. Shape Correspondence

To compare two shapes $\mathcal{M}$ and $\widetilde{\mathcal{M}}$ with each other, we need two ingredients. First, we employ a feature that will describe our shape pointwise well enough to make comparison possible. Second, we need a metric so we can declare which features are close to each other and which are not.

#### 2.2.1. Feature Descriptor

For establishing shape correspondence, we consider for each point on an object's surface a feature descriptor, which is a computational object that contains geometric shape information. To meet this aim, we will solve the wave or damped wave equation on the shape, respectively. Depending on the starting point $x_i$ and the local geometry around $x_i$,

the solutions $u(x,t)$ of these PDEs will propagate differently. The feature descriptor of $x_i$ will be the solution $u(x,t)$ restricted to the spatial component

$$f_{x_i}(t) := u(x,t)|_{x=x_i} \tag{3}$$

Roughly speaking, we expect that the feature descriptors of different points differ enough to not get mixed up with each other but are quite similar for matching points on variations of the same shape (undergoing almost isometric transformations).

To illustrate the idea about the nature of the feature descriptors we compute, we show in Figure 2 the analytical solution $u(x,t)$ of the damped wave equation for a cosine initial condition propagating in time. The feature descriptor $f_{x_i}$ is plotted as the black line in this figure. For more details, we refer to Section 5.

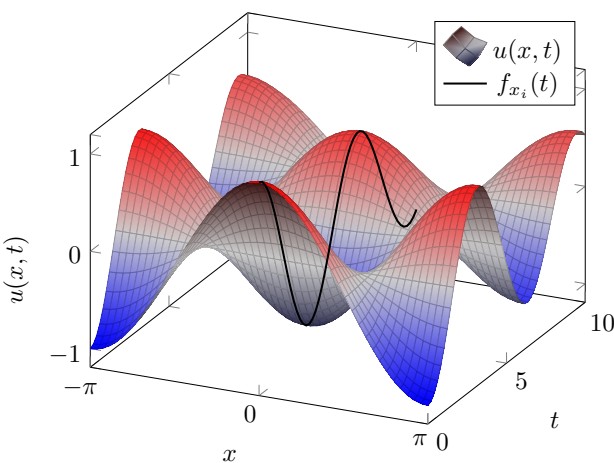

**Figure 2.** The one-dimensional analytical solution of the damped wave equation. The spatial interval is chosen from $-\pi$ to $\pi$ and $[0,10]$ for the time interval. The feature descriptor $f_{x=0}$ is the solid black line and is the function $u(x,t)$ restricted to $x=0$. For more details on the solution and how we obtain it, we refer the reader to Section 5.

In our framework, we need to compute the feature descriptor of all points on the different shapes. After that, we compare them with each other and search for the best matching ones.

### 2.2.2. Metric

To compare the feature descriptors for different locations $x_i \in \mathcal{M}$ and $\tilde{x}_j \in \widetilde{\mathcal{M}}$ on two different shapes $\mathcal{M}$ and $\widetilde{\mathcal{M}}$, we define a distance $d_f(x_i, \tilde{x}_j)$ via the $L_1$ norm

$$d_f(x_i, \tilde{x}_j) = \int_{\mathcal{I}} |f_{x_i} - f_{\tilde{x}_j}| \, \mathrm{d}t \tag{4}$$

over the interval $\mathcal{I} = [0,T]$. This will compute the absolute difference between two feature descriptors. At the end, the tuple of locations $(x_i, \tilde{x}_j) \in \mathcal{M} \times \widetilde{\mathcal{M}}$ with the smallest distance are considered to belong together.

### 2.3. PDE-Based Models

Since we consider a PDE for generating the feature descriptor, the choice of a good PDE to meet this purpose is a core question of the shape-matching framework.

In this paper, we want to focus on the wave and damped wave equation, respectively. Other PDEs such as the heat equation [4] have already been discussed for our framework to some extent [7,9,10].

The classical wave and damped wave equation make use of the Laplace operator $\Delta$. When we operate on $\mathcal{M} \subset \mathbb{R}^2$, we can restrain the Laplace operator to the local variables of

our manifold. The corresponding operator is called Laplace-Beltrami operator (LBO) $\Delta_{\mathcal{M}}$, and it will respect the curvature of the manifold. Applied to a scalar-valued function $u$, we can write it as:

$$\Delta_{\mathcal{M}} u = \frac{1}{\sqrt{|g|}} \sum_{i,j=1}^{2} \partial_i \left( \sqrt{|g|} g^{ij} \partial_j u \right) \tag{5}$$

where $|g|$ is the determinant of the metric tensor $g \in \mathbb{R}^{2 \times 2}$ that describes locally the geometry, and $g^{ij}$ are the entries of its inverse; see, e.g., [11] for more details on the differential geometric notions. The LBO itself is related to the mean curvature of a shape [12]. Shape representations making use of the Gaussian curvature are also possible [13,14].

### 2.3.1. Wave Equation and Damped Wave Equation

Let us first recall the *wave equation* introduced for the shape matching scenario in [6],

$$\partial_{tt} u(x, t) = \Delta_{\mathcal{M}} u(x, t), \ x \in \mathcal{M}, \ t \in \mathcal{I} \tag{6}$$

on a surface $\mathcal{M}$ and the interval $\mathcal{I} = [0, T]$. As has been shown in [6], this PDE may be considered the PDE-based model that gives the best results among the PDEs from classical physics in the numerical shape-matching setting. As a key model for our investigations, we employ the *damped wave equation*

$$\partial_{tt} u(x, t) + k \partial_t u(x, t) = \Delta_{\mathcal{M}} u(x, t) \tag{7}$$

with $x \in \mathcal{M}$ and $t \in \mathcal{I}$. Choosing $k = 0$ will lead back to the standard wave Equation (6). Making use of both equations, we study the expansion of an initial wave over the surface of a given shape.

To distinguish clearly between the corresponding feature descriptors, we employ the following notions. We use the *damped feature descriptor* to denote the feature descriptor obtained from the solution of the damped wave Equation (7) and just *feature descriptor* to denote the one obtained from the wave equation without damping (6).

### 2.3.2. Initial Conditions

Both considered PDEs need to be supplemented by an initial function and an initial velocity to solve them properly. In all our scenarios, we will set the initial velocity $\partial_t u(x, 0) = 0$. Let us now comment on the chosen initial functions.

First, we use the Dirac delta function

$$u(x, 0) = u^{\delta}(x) = \delta_{x_i}(x) = \begin{cases} 1, & x = x_i \\ 0, & x \neq x_i \end{cases} \tag{8}$$

centred at the point $x_i \in \mathcal{M}$; cf. [4] for use of this function in the context of feature computation with PDE.

The second initial function we use is a highly peaked Gaussian distribution

$$u(x, 0) = u^G(x) = \exp \left( -\frac{1}{2} \frac{d_{\mathcal{M}}^2(x, x_i)}{\sigma^2} \right) \tag{9}$$

around the centre $x_i \in \mathcal{M}$ and with $\sigma^2$ as the width parameter. Letting $\sigma$ converge towards zero would lead to the Delta distribution:

$$\lim_{\sigma \to 0} \exp \left( -\frac{1}{2} \frac{d_{\mathcal{M}}^2(x, x_i)}{\sigma^2} \right) = \delta_{x_i}(x) \tag{10}$$

### 3. Basic Discretisation

Let us recall the basic discretisation, as it has already been employed, for instance, in [6]. The discrete shape representation $\mathcal{M}_d = (\mathcal{P}, \mathcal{T})$ of our shape $\mathcal{M}$ can be realised through a set of points $\mathcal{P} := \{x_1, \ldots, x_N\}$ containing coordinates of each point $x_i$ and a set of triangles $\mathcal{T}$. In $\mathcal{T}$, we store $G$ triplets $i, j, k$, which indicate that the points $x_i, x_j, x_k \in \mathcal{P}$ form a triangle together. Furthermore, we denote by $\Omega_i$ the barycentric cell volume surrounding the point $x_i$.

In Figure 3, we visualise these terminologies. We plotted the shape of the centaurs head on the left. In the middle, we show the discrete version of this head. On the right image, we zoomed in on the discrete model to show the points, angles and barycentric cell volume.

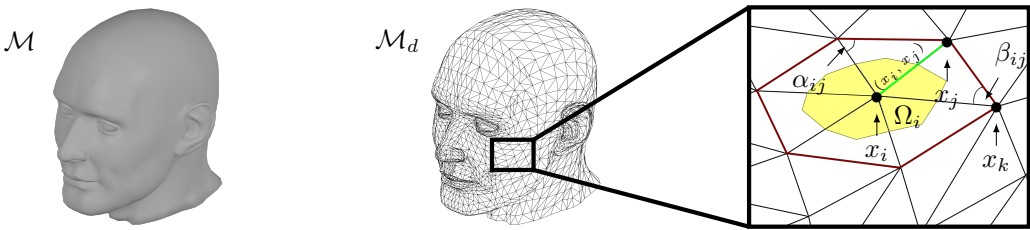

**Figure 3.** (**Left**) The 2D Riemannian manifold $\mathcal{M}$ of the centaur's head. (**Middle**) The discrete approximation $\mathcal{M}_d$ of this manifold. $\mathcal{M}_d$ is displayed via the point cloud $\mathcal{P}$ and the triangles $\mathcal{T}$. (**Right**) The setting for the calculation of the LBO around the point $x_i$. The yellow area indicates the cell volume $\Omega_i$. Additionally, we see the two angles $\alpha_{ij}$ and $\beta_{ij}$ opposite to the edge $(x_i, x_j)$ (green line).

#### 3.1. Spatial Discretisation

We now want to explain spatial discretisation. Let us illustrate the proceeding at hand of Equation (7). Setting $k = 0$ in this example will get us the discretisation for Equation (6).

First, we consider Equation (7) over a shape $\mathcal{M}$ and time interval $\mathcal{I}$. The integration over time and space will lead to

$$\int_{\mathcal{I}} \int_{\mathcal{M}} \partial_{tt} u(x, t) + k \partial_t u(x, t) \, \mathrm{d}x \, \mathrm{d}t = \int_{\mathcal{I}} \int_{\mathcal{M}} \Delta_{\mathcal{M}} u(x, t) \, \mathrm{d}x \, \mathrm{d}t \tag{11}$$

In the discretisation, we subdivide our meshed surface $\mathcal{M}_d$ and the time interval $\mathcal{I}$ in the following manner:

$$\mathcal{M}_d = \bigcup_{i=1}^{N} \Omega_i, \quad \mathcal{I} = \bigcup_{l=1}^{M} \mathcal{I}_l \quad \text{with} \quad \mathcal{I}_l = [t_{l-1}, t_l] \tag{12}$$

with $N$ the total number of barycentric cell volumes $\Omega_i$ located around the point $x_i$ and $M$ the total number of time intervals $0 := t_0 < t_1 < \ldots < t_M := T$ with the time increment $\tau = t_l - t_{l-1}$ uniform for all $l \in \{1, \ldots, M\}$. This allows us to focus on one arbitrarily chosen time period with $t \in \mathcal{I}_l$ and a fixed cell $\Omega_i$:

$$\int_{\mathcal{I}_l} \int_{\Omega_i} \partial_{tt} u(x, t) + k \partial_t u(x, t) \, \mathrm{d}x \, \mathrm{d}t = \int_{\mathcal{I}_l} \int_{\Omega_i} \Delta_{\mathcal{M}} u(x, t) \, \mathrm{d}x \, \mathrm{d}t \tag{13}$$

Additionally, with the definition of the cell average

$$u_i(t) = \frac{1}{|\Omega_i|} \int_{\Omega_i} u(x, t) \approx u(\bar{x}_i, t) \, \mathrm{d}x \tag{14}$$

where the approximation is an identity up to second-order accuracy [15] and where $|\Omega_i|$ denotes the area of $\Omega_i$, we can calculate the averaged Laplacian as

$$Lu_i(t) = \frac{1}{|\Omega_i|} \int_{\Omega_i} \Delta_{\mathcal{M}} u(x,t) \, dx \tag{15}$$

This integral is solved with the use of the divergence theorem and the application of the cotangent weight scheme [16], which we will recall in a moment. In the end, we obtain the ODE system

$$\ddot{\mathbf{u}}(t) + k\dot{\mathbf{u}}(t) = L\mathbf{u}(t) \tag{16}$$

where the $N$-dimensional vector $\mathbf{u}(t) = (u_1(t), \ldots, u_N(t))^\top$ will store all functions $u_i(t)$.

The cotangent weight scheme will give us the discrete LBO $L \in \mathbb{R}^{N \times N}$ [17,18]. This matrix is composed via the sparse matrix representation $L = D^{-1}W$ with

$$W_{ij} = \begin{cases} -\sum\limits_{j \in \nu_i} w_{ij}, & \text{if } x_i = x_j \\ w_{ij}, & \text{if } x_i \neq x_j \text{ and } j \in \nu_i \\ 0, & \text{else} \end{cases} \tag{17}$$

computed with the cotangent formulae

$$w_{ij} = \begin{cases} \frac{\cot \alpha_{ij} + \cot \beta_{ij}}{2} & \text{if } (x_i, x_j) \in E_i \\ \frac{\cot \alpha_{ij}}{2} & \text{if } (x_i, x_j) \in E_b \end{cases} \tag{18}$$

Thereby, $\nu_i$ denotes the neighbourhood of vertexes $x_i$, $E_i$ as the set of interior edges and $E_b$ as the set of boundary edges. Let us mention that in our considered scenarios, $E_b = \varnothing$, since we assume we deal with a closed manifold. The angles $\alpha_{ij}$ and $\beta_{ij}$ are opposite to the edge $(x_i, x_j)$, and the matrix $D = \text{diag}(|\Omega_1|, \ldots, |\Omega_i|, \ldots, |\Omega_N|)$ contains the local cell areas. A visualization of this can be seen in the right plot of Figure 3.

In order to enforce a uniform numerical proceeding with respect to all occurring time derivatives, we want to transform the Equation (16) into an ODE system of first-order. We obtain

$$\dot{\mathbf{q}}(t) = H\mathbf{q}(t) \tag{19}$$

due to the choice of $\mathbf{q}(t) = (\mathbf{u}(t), \dot{\mathbf{u}}(t))^\top \in \mathbb{R}^{2N}$ and

$$H = \begin{pmatrix} 0 & I_N \\ L & -kI_N \end{pmatrix} \in \mathbb{R}^{2N \times 2N} \tag{20}$$

with $I_N \in \mathbb{R}^{N \times N}$ being the identity matrix.

At this point, it is adequate to talk about the discrete initial conditions. As in the analytical scenario, we set the initial velocity to the null vector $\dot{\mathbf{u}}(0) = \mathbf{0}$. The discrete versions of the initial conditions (8) and (9) can be written as

$$\mathbf{u}_{x_i}^\delta = \mathbf{u}(x_i, 0) = (0, \ldots, 0, |\Omega_i|^{-1}, 0, \ldots, 0)^\top \tag{21}$$

and

$$\mathbf{u}_{x_i}^G = |\Omega_i|^{-1}(u^G(x_1), \ldots, u^G(x_i), \ldots, u^G(x_N))^\top \tag{22}$$

### 3.2. Eigenproblem and Modal Coordinate Reduction

Systems such as (16) or (19) come with large sparse matrices, and thus their repeated solution consequently leads to high costs in terms of computation time. In order to approach this issue, one of the natural ideas is to reduce the order of the matrices from $N$, such as in Equation (16) or $2N$ such as Equation (19), to a smaller order $r$. The Modal Order Reduction

(MOR) technique will offer us a suitable framework for performing this in a suitable way. With this technique, a low-dimensional system will supplant the current high-dimensional one. We recommend the reader to [19,20] for more details on the MOR topic.

The MOR technique itself includes a large variety of methods. Specifically, in our framework, we want to focus on the Modal Coordinate Reduction (MCR) technique. In [9], an approach was presented, which we will briefly recall now.

The key step is the calculation of the eigenvalues $\lambda$ and eigenvectors $\mathbf{v}$ of the LBO $L\mathbf{v} = \lambda\mathbf{v}$. With $L = D^{-1}W$, we may reformulate this into the generalised eigenvalue problem

$$W\mathbf{v} = \lambda D\mathbf{v} \tag{23}$$

The symmetry of $W$ and the diagonal matrix $D$ will lead to real eigenvalues and linear independent eigenvectors. Additionally, we determine that the eigenvectors $\mathbf{v}$ are $D$-orthogonal with $\mathbf{v}_i^\top D\mathbf{v}_j = \delta_{ij}$. These considerations will lead to the following equations:

$$I = V^\top DV, \quad L = V\Lambda V^\top D, \quad \Lambda = V^\top WV \tag{24}$$

with $V$ being the right eigenvector matrix of $L$, and $\Lambda$ being the diagonal matrix of eigenvalues. Multiplying the second equation with $V$ from the right will give the basic property $LV = V\Lambda$, making use of the first identify in (24). The third equation can be obtained via the multiplication of $V^\top D$ from the left on $LV = D^{-1}WV = V\Lambda$.

To finally achieve the MCR transformation, we use these considerations and substitute Equation (16) with $\mathbf{u} = V\mathbf{w}$, leading to

$$V\ddot{\mathbf{w}} + kV\dot{\mathbf{w}} = LV\mathbf{w} \tag{25}$$

With the left multiplication of $V^\top D$, we obtain

$$\ddot{\mathbf{w}} + k\dot{\mathbf{w}} = \Lambda\mathbf{w} \tag{26}$$

Let us mention that, at this point, we still have a high-dimensional system since we consider all eigenvalues. The reduction happens when we use the first $r \ll N$-ordered eigenvalues $0 = \lambda_1 < \lambda_2 \leq \ldots \leq \lambda_r$. The reduced model writes as

$$\ddot{\mathbf{w}}_r + k\dot{\mathbf{w}}_r = \Lambda_r\mathbf{w}_r \quad \text{where} \quad \mathbf{w}_r = V_r^\top D\mathbf{u} \tag{27}$$

Analogously to the steps from (16) to (19), we produce from (27) the equation

$$\dot{\mathbf{p}}_r(t) = H_r\mathbf{p}_r(t) \quad \text{with} \quad H_r = \begin{pmatrix} 0 & I_r \\ \Lambda_r & -kI_r \end{pmatrix} \in \mathbb{R}^{2r\times 2r}, \quad \mathbf{p}_r = (\mathbf{w}_r, \dot{\mathbf{w}}_r)^\top \in \mathbb{R}^{2r} \tag{28}$$

The last step remaining is the transformation of the initial conditions into our reduced system. Beginning with $\mathbf{u}^\delta(x_i, 0) = (0, \ldots, 0, |\Omega_i|^{-1}, 0, \ldots, 0)^\top$, we obtain with $\mathbf{w}_r^\delta = V_r^\top D\mathbf{u}^\delta$ the reduced condition $\mathbf{w}_r^\delta(x_i, 0) = V_r^\top \mathbf{e}_i$. For the Gaussian initial condition, we receive with $\mathbf{w}_r^G = V_r^\top D\mathbf{u}^G$

$$\mathbf{w}_r^G(x_i, 0) = V_r^\top \left( u^G(x_1), \ldots, u^G(x_i), \ldots, u^G(x_N) \right)^\top \tag{29}$$

The initial velocity $\dot{\mathbf{w}}_r(0) = \mathbf{0}$ will not change, since it was already zero.

### 3.3. Improvement of the Eigenvalue Computation

Since the main aspect of the MCR method is the computation of the eigenvalues $\lambda$ and eigenvectors $\mathbf{v}$, we want to show an improvement of the eigendecomposition, as mentioned

in [9]. We want to transform our problem into the symmetric standard eigenvalue problem via the similarity transformation.

$$L = D^{-1}W = D^{-\frac{1}{2}}D^{-\frac{1}{2}}WD^{-\frac{1}{2}}D^{\frac{1}{2}} = D^{-\frac{1}{2}}BD^{\frac{1}{2}} \tag{30}$$

with the symmetric, inherited from matrix $W$, matrix $B = D^{-\frac{1}{2}}WD^{-\frac{1}{2}}$. Due to this transformation, the matrices $L$ and $B$ are similar and share the same real eigenvalues. However, the eigenvectors of $L$ and $B$ differ. With $\tilde{\mathbf{v}}$, we denote the eigenvectors of $B$, and $\mathbf{v}$ refers to the eigenvectors of $L$. Then, $\mathbf{v}$ and $\tilde{\mathbf{v}}$ can be transformed into each other via

$$\mathbf{v} = D^{-\frac{1}{2}}\tilde{\mathbf{v}}, \quad \tilde{\mathbf{v}} = D^{\frac{1}{2}}\mathbf{v} \tag{31}$$

*3.4. Time Discretisation*

In our experiments, we use three different numerical schemes for time discretisation. We have two $l_0$-stable schemes, namely the implicit Euler (IE) method as a method of first-order accuracy in time and then a method of second-order presented in [21]. The third method we use is the Crank-Nicolson (CN) method as a method of second-order accuracy, which is not $l_0$-stable. We showed in our previous work [10] that $l_0$-stability is an important property in the shape-matching framework and should not be left out.

As already mentioned, we subdivided the time interval $\mathcal{I} = [0, t_M]$ into smaller intervals $\mathcal{I}_l = [t_{l-1}, t_l]$ with $0 = t_0 < t_1 < \ldots < t_M$. Additionally, we choose uniform time increments $\tau = t_l - t_{l-1}$.

As discussed in [9] we change the temporal domain $[0, t_M]$ for wave and damped wave equations to $[0, t^*]$ using

$$t^*(\lambda_r) = \frac{t_M \sqrt[4]{|\lambda_N|}}{\sqrt[4]{|\lambda_r|}} \tag{32}$$

The computational parameters are thus $\tau = \frac{t^*}{M}$, with $M$ being the number of iterations.

3.4.1. Implicit Euler and Crank–Nicolson

To solve the time derivatives, we integrate the differential Equation (28) over $\mathcal{I}_l$ either with an approximation by the rectangle method to obtain the IE scheme

$$\mathbf{p}_r(t_l) - \mathbf{p}_r(t_{l-1}) = \int_{t_{l-1}}^{t_l} H_r\mathbf{p}_r(t)\,\mathrm{d}t \approx \tau H_r\mathbf{p}_r(t_l) \tag{33}$$

or the trapezoidal rule to obtain the CN scheme

$$\mathbf{p}_r(t_l) - \mathbf{p}_r(t_{l-1}) = \int_{t_{l-1}}^{t_l} H_r\mathbf{p}_r(t)\,\mathrm{d}t \approx \frac{\tau}{2}H_r(\mathbf{p}_r(t_{l-1}) + \mathbf{p}_r(t_l)) \tag{34}$$

With some minor transformations, we obtain

$$\mathbf{p}_r(t_l) = (I_{2r} - \tau H_r)^{-1}\mathbf{p}_r(t_{l-1}) \tag{35}$$

for the IE method and

$$\mathbf{p}_r(t_l) = \left(I_{2r} - \frac{\tau}{2}H_r\right)^{-1}\left(I_{2r} + \frac{\tau}{2}H_r\right)\mathbf{p}_r(t_{l-1}) \tag{36}$$

for the CN method.

3.4.2. Second-Order Time Integration

First-order accurate methods are easier to construct than the majority of second-order methods, which may be produced by relying on Butcher tableaus. These tableaus may become quite complicated when the scheme has to also be $l_0$-stable.

In this section, we recall a second-order $l_0$-stable (SO$l_0$) scheme, first presented in [21], that does not rely on Butcher tableaus and is relatively easy to compute. In the following, we want to give a brief explanation of this method.

We aim to solve an equation like (19) or (28), and these equations look like

$$\dot{\mathbf{x}}(t) = A\mathbf{x}(t) \tag{37}$$

The solution of this ODE is $\mathbf{x}(t) = \exp(tA)$, and for a time step further, we can write

$$\mathbf{x}(t + \tau) = \exp((t + \tau)A) = \exp(\tau A)\mathbf{x}(t) \tag{38}$$

However, calculating $\exp(\tau A)$ is a complicated task. To simplify it, we make use of approximations such as

$$R(\tau A) \approx \exp(\tau A) \tag{39}$$

A simple choice of $R(\tau A)$ would be $R(\tau A) = (I - a\tau A)^{-1}(I + (1 - a)\tau A)$, which leads to well-known numerical schemes just by changing $a$. Choosing $a = 0$, we obtain the explicit Euler method. For $a = 1$, we obtain the IE method, as it is presented earlier in this paper. The CN scheme can be produced by setting $a = 0.5$.

To obtain a $l_0$-stable second-order method, we need another $R(\tau A)$. Twizell et al. presented the following equation

$$R(\tau A) = (I - r_1\tau A)^{-1}(I - r_2\tau A)^{-1}(I + (1 - a)\tau A) \tag{40}$$

as such a suitable choice. With $r_{1,2} = \frac{1}{2}\left(a \mp \sqrt{a^2 - 4a + 2}\right)$, we obtain a method of second-order, and by choosing $a = 2 - \sqrt{2} - \varepsilon$, with $\varepsilon$ being an arbitrarily small positive number, we obtain a $l_0$-stable method which relies only on real-valued arithmetic.

Putting these considerations on our Equation (28) leads to the following numerical scheme:

$$\mathbf{p}_r(t_l) = (I_{2r} - r_1\tau H_r)^{-1}(I_{2r} - r_2\tau H_r)^{-1}(I_{2r} + (1 - a)\tau H_r)\mathbf{p}_r(t_{l-1}) \tag{41}$$

**4. Experimental Settings**

After presenting all the mathematical background, we now turn to basic matters of implementation and testing. With this aim, we present detailed information about the data set we used, the way we declare whether two given shapes belong together or not, and, last but not least, we provide a flowchart of our code.

*4.1. Dataset*

We used a selection of the classic TOSCA data set [2]. This set has a total of 80 shapes from animals and humans in different positions. The resolution of these shapes goes from 4,344 points up to 52,565 points. The shapes from this data set have a ground-truth map. Thus, for overall shapes of one and the same kind, we know which points belong together.

Unless we say otherwise, we used the centaur shape (cf. Figure 1) as our main shape for all the experiments. We chose this shape because it has both human and animal parts. For the sake of completeness, we would like to mention that the centaur consists of $N = 15,768$ points.

*4.2. Evaluation and Reference Models*

When do we know that of two shapes, one is obtained by an almost isometric transformation of the other shape? To answer this question, we would like to follow the Princeton

benchmark protocol [22]. For this, we need to explain the concepts of the geodesic error and the hit rate.

The geodesic error is an important step in the Princeton benchmark protocol for calculating the correspondence quality. To this end, we determine the normalised intrinsic distance $d_{\mathcal{M}}(x_i, x^*)/\sqrt{A_{\mathcal{M}}}$ between the computed match $x_i \in \mathcal{M}$ and the ground-truth $x^*$. With $A_{\mathcal{M}}$, we declare the total area of the shape.

With this definition, we effectively draw a circle around the matching point $x_i$ with a radius of a threshold parameter. Starting at 0, we increase the radius until the ground-truth $x^*$ is inside the circle, but not more than 0.25.

If $x^*$ is inside the circle with a radius of at least 0.25, this matching is declared as true and false otherwise. Spoken in these terms, we have a perfect matching when we have a threshold parameter of 0.

The hit rate defines the rate of true matchings and can be calculated with

$$\text{hit rate} = \frac{\text{TP}}{(\text{TP} + \text{FN})} \tag{42}$$

With TP, we describe the number of true positive results, and FN describes the number of false negative results.

*4.3. Our Code*

At this point in the paper, we want to explain the basic steps of the code used for the shape matching calculations. All steps of Algorithm 1 refer to the equations explained within this paper.

---

**Algorithm 1:** Shape Matching MOR Method

**Data:** Shapes
**Result:** hit rate between shapes

1 **foreach** *Shape* **do**
2     Compute $W$ and $D^{-1}$ for the LBO via (17) and (18).;
3     Compute eigendecomposition $(\lambda, \tilde{\mathbf{v}})$ of $B$ (30) to get the eigenvectors of the LBO $L$ with (31);
4     Keep the $r$ lowest nonzero eigenvalues $\lambda_r$ and the corresponding eigenvectors $V_r$. Additionally, keep the maximal eigenvalue $\lambda_N$ for the temporal domain calculation;
5     Compute temporal domain $t^*$ with (32) and the time increment $\tau = t^*/M$;
6     Compute $H_r$ via (28);
7     **foreach** $x \in \mathcal{P}$ **do**
8         Set initial condition $\mathbf{u}(x_i, 0)$ with (8) or (9);
9         Transform the initial condition into the reduced system $\mathbf{w}_r(x_i, 0)$ via (27);
10         Solve this with one of our three models (35), (36) or (41);
11         Transform the solution back into the full system via $\mathbf{u} = V_r \mathbf{w}$;
12         Extract the damped feature descriptor $f_{x_i}(t)$ from $\mathbf{u}$ (3);
13     **end**
14 **end**
15 Compare feature descriptors of different shapes (4) to find the best matching ones;
16 Compute the hit rate using Equation (42);

---

Overall, this code does not need much input. A basic version of the algorithm needs the shapes and two parameters, the number of the eigenvalues to keep for modal coordinate reduction $r$, and the end of the time interval $t_M$. Both parameters can be set to $r = t_M = 100$. That this is a suitable choice for the parameters has been verified in previous works [6,7,9].

If we want to use the damped wave equation or the Gaussian initial conditions, we have to pass the damping parameter $k$ and the width of the Gaussian function $\sigma$ as well.

## 5. Introductory Synthetic Tests

We now conduct in a first experimentally oriented section a series of comparatively simple, mainly synthetic experiments. These experiments are aimed to show very clearly important computational aspects and they allow us to perceive these independently from potentially perturbing parameters, such as, e.g., varying shape resolution or noise in a shape's acquisition.

Let us nevertheless start discussing a given example shape, serving as a motivation for the considerations. Running a basic experiment on the centaur shape, with delta peak initial condition (8) and the three methods to solve the wave Equation (6), we notice that the second-order methods perform very different from the first-order method. In order to see this, let us refer to Figure 4 where we show the feature descriptors from different points on the shape. One point is on the left thumb, one on the back and one at the hoof. With the blue circles and red triangles, we distinguish between Shape 0 and Shape 2, or left and right in Figure 1. In the rows, we see the different numerical schemes and the columns indicate the different points. We notice that the feature descriptors of the second-order methods will result in jagged curves. On the other hand, the first-order method will lead to smooth curves.

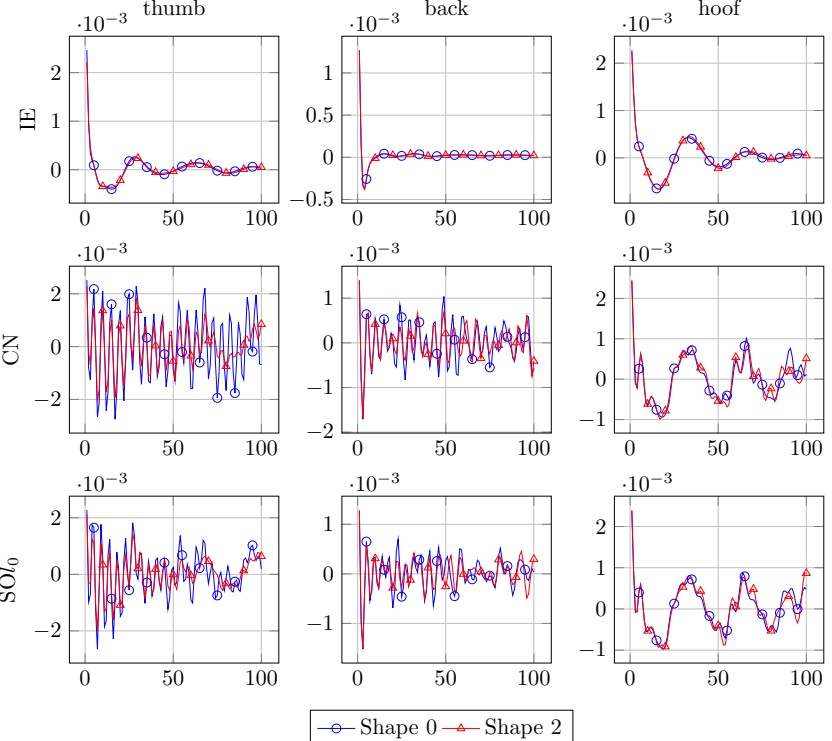

**Figure 4.** Visualisation of the feature descriptor at different points of the centaur shape, obtained through the solution of the wave equation using the IE solver with $\delta$-peak initial conditions. Shape 0, Shape 2 and the point are equal to the points, and shapes are shown in Figure 1. (**Rows**) Different solver. (**Columns**) Different points.

Naturally, however, one would expect that all the schemes give an account of the underlying analytical model, where the second-order methods are expected to give a more accurate solution than the first-order scheme. Thus, we conjecture that we observe some hidden computational parameters that have an important impact on feature construction.

In this section, we want to explain these observations, and in doing so, we attempt to find a better model for the shape-matching scenario. To pursue this aim, we study an academic example. Furthermore, we can also explain in some detail the idea in the use of the damped wave equation as a general model for some important phenomena in the feature descriptor computation.

### 5.1. Academic Examples

When we want to study our model properties with the numerical schemes, it comes in handy to find an analytical solution for the feature descriptor, first. For this, we need a more practicable setting than the centaur shape.

We will use the 1D damped wave equation $u_{tt} + 2ku_t = u_{xx}$ on the interval of $x \in \left[-\frac{L}{2}, \frac{L}{2}\right]$, periodic boundary conditions and initial velocity $u_t(x, t) = 0$, for this purpose. The value 2 in this equation will make the calculation of the analytical solution easier to execute. In the following, we will outline the extraction of the solution, for the sake of completeness. In doing so, we want to be as general as possible and as detailed as necessary.

With the method of separation of variables and the condition,

$$1 < \frac{2\pi}{kL} \tag{43}$$

we receive the analytical solution $u(x, t) = \sum_n u_n(x, t)$ via

$$u_n(x, t) = e^{-kt} A_n \left( \cos(\alpha_n t) + \frac{k}{\alpha_n} \sin(\alpha_n t) \right) \cos(\beta_n x) \tag{44}$$

The condition (43) is needed to generate an oscillating solution that may give a meaningful result for generating a feature. Furthermore, we set $\alpha_n = \sqrt{\beta_n^2 - k^2}$ and $\beta_n = \frac{2n\pi}{L}$. In the standard shape-matching scenario, we would use $u^\delta(x)$ or $u^G(x)$ as initial conditions $u(x, 0)$. Since we would have to Fourier transform these initial conditions, and this will not make the solution more pleasant, we choose the cosine function $u(x, 0) = \cos(x)$ as the initial condition instead. With this choice and $L = 2\pi$, we can determine the parameter $A_n$ in Equation (44):

$$\cos(x) = u(x, 0) = \sum_n A_n \cos(nx) \tag{45}$$

which implies

$$A_n = \begin{cases} 1 & \text{if } n = 1 \\ 0 & \text{if } n \neq 1 \end{cases} \tag{46}$$

In the end, we get the analytical solution for our academic example

$$u(x, t) = e^{-kt} \left( \cos(\sqrt{1 - k^2} t) + \frac{k}{\sqrt{1 - k^2}} \sin(\sqrt{1 - k^2} t) \right) \cos(x) \tag{47}$$

From this equation, by choosing a specific $x$, we can also obtain the feature descriptor. We choose the centre of the domain $x = 0$, meaning

$$f_{x=0}(t) = u(x, t)|_{x=0} = e^{-kt} \left( \cos(\sqrt{1 - k^2} t) + \frac{k}{\sqrt{1 - k^2}} \sin(\sqrt{1 - k^2} t) \right) \tag{48}$$

Setting $k = 0$ will generate the solution and feature descriptor for the wave equation. As we did earlier, we will write *analytical (damped) feature descriptor* for the feature descriptor obtained from the analytical solution of the (damped) wave equation.

In Figure 2 we already presented the solution $u(x, t)$ and the feature descriptor of this example. The solution $u(x, t)$ is presented as a red-blue surface. Reddish parts are greater than zero, and bluish parts are less than zero. Additionally, we plotted the feature descriptor $f_0$ as a solid black line. For the evaluation, we have chosen the time increment $\tau = 0.1$ and the damping constant $k = \frac{\tau}{2}$.

### 5.2. Experiments

Now, after explaining our academic setting, we want to use this to study the parameters for a good shape-matching model. In addition, we want to present insights into the use

of the damped wave equation and Gaussian initial condition. Thereby, we want to explain the jagged curves that occur in the feature descriptors of the second-order methods.

### 5.2.1. Details for the Damped Wave Equation

We already noticed in our last work [10] that the second-order methods CN and $SOl_0$ did not perform favourably compared to the first-order method IE.

Using numerical solvers for PDEs will always result in a solution with an approximation error compared to the exact solution. We could, in principle, attempt to conduct an analytical error analysis of the numerical schemes to determine accounts of the wave propagation and damping parts in the leading order errors, but this is actually a quite tedious issue for the $SOl_0$ scheme, which is of importance in our numerical study. As it turns out, when approaching this task, the resulting terms are not as clear to interpret as desirable. It is even more challenging and a great deal beyond the scope of this study to perform a theoretical error analysis for the setting on a shape.

Thus, we opt to present here a more vivid approach that clearly illustrates the underlying mechanisms. Since we have the analytical damped feature descriptor (48), we can adjust the damping parameter $k$ in a way that it will fit the feature descriptors obtained from one of the numerical solvers. For the evaluation, we chose $\tau = 0.1$ as the time increment and $h = 0.05$ as the lattice parameter.

In Figure 5, we present the results of our approach. In this figure, the red circles indicate the analytical feature descriptor obtained from the IE method. The blue triangles belong to the CN scheme, the solid black line is the analytical damped feature descriptor, and the solid red line shows the difference between the second-order methods.

In the top row in Figure 5, we changed the damping parameter $k$ from the analytical damped feature descriptor to fit best with the feature descriptors from the IE (left) or the CN method (right). For the fit to the IE method, we used a damping parameter of $k = \tau/2$. In the top right plot, we used $k = \tau/8$ as the damping parameter. This indicates that the numerical solution of a wave equation obtained via the IE method is closer to the analytical solution of the damped wave equation than to the analytical solution of the wave equation. This phenomenon is well known under the topic of modified equations [23].

In the bottom right plot, we showed that it is possible to imitate the analytical feature descriptor from the IE method with the analytical damped feature descriptor from the CN method. The damping parameter was set to $k = \tau$.

In total, this means that the solution of the wave equation using the IE method and the solution of the damped wave equation using the CN method are approximately the same. This leads us to conjecture that it is possible, with an adequate choice of the damping parameter, to describe beneficial properties of the IE method in an analytical way. This may give an indication about a potentially necessary (or beneficial) damping behaviour in continuous-scale PDE models that may be useful in a more general shape matching scenario.

In the bottom left picture of Figure 5, we present the difference between the two second-order methods. We observe that the difference becomes slightly larger during time evolution, but for our time interval, they are close to each other. This result leads us to plot only the CN method as representative of the second-order methods.

The error between the second-order methods will grow, and at a certain point, the results of both schemes are not equal anymore. To show this, we set up Figure 6. In this plot, we see the cosine initial condition at the top and the second-order schemes at different iterations on the bottom. We have chosen iterations close to the 100th, 250th and 500th iteration to show the differences between both solutions. We chose iterations close to the mentioned ones because we want to show the turning point for the $SOl_0$ method. In doing so, the comparison to the CN gets easier. Both solutions will simply oscillate up and down without any movement to the sides. The blue line belongs to the CN scheme, and the green one belongs to the $SOl_0$ method. For obtaining faster results, we increased the time increment from $\tau = 0.1$ to $\tau = 0.5$.

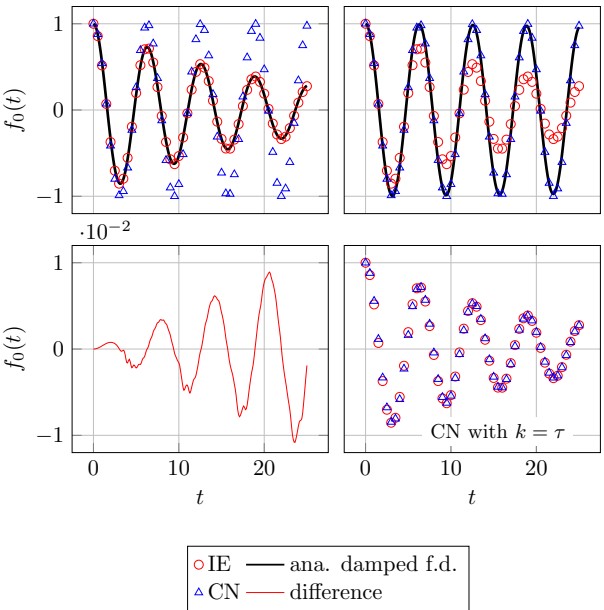

**Figure 5.** For all plots, red circles indicate the CN method, and the blue triangle denotes the $\text{SO}l_0$ scheme. The black solid line indicates the analytical damped feature descriptor, and the red line stands for the difference between the two analytical damped feature descriptors obtained from the second-order methods. (**Top left**) The damping parameter for the analytical damped feature descriptor was set to $k = \tau/2$. (**Top right**) Damping parameter was set to $k = \tau/8$. (**Bottom left**) We plotted the difference between the analytical damped feature descriptors obtained with the CN and $\text{SO}l_0$ method. The difference will not exceed $\pm 1 \cdot 10^{-2}$. Therefore, only CN is used as a representative of the second-order methods. (**Bottom right**) We compared the damped feature descriptor ($k = \tau$) obtained using the CN method with the feature descriptor using IE. All (damped) feature descriptors are extracted from a solution $u(x, t)$ at the point $x = 0$. For the numerical calculations, we set the time increment $\tau = 0.1$ and the lattice parameter to $h = 0.05$.

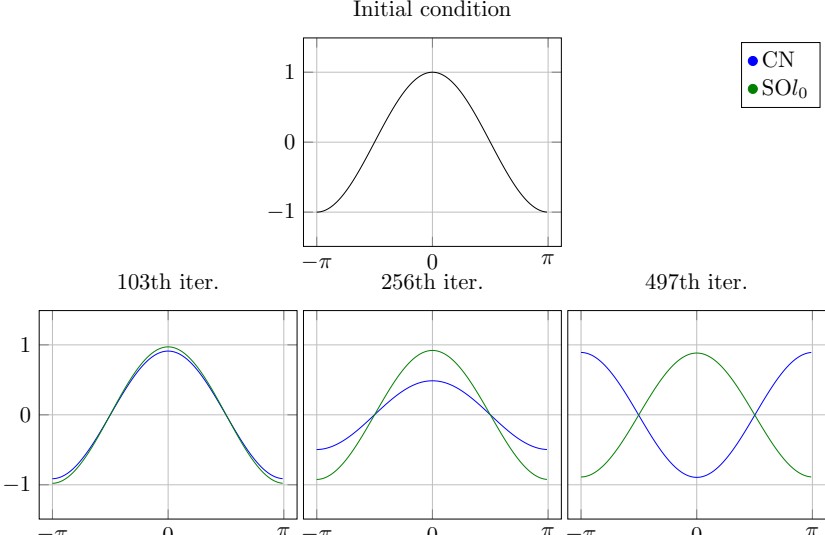

**Figure 6.** (**Top row**) Cosine initial condition. (**Bottom row**) We plotted the solution from our academic setting at different iterations. The solution was generated using the $\text{SO}l_0$ (green) scheme and the CN (blue) method. We chose iterations close to 100, 250 and 500 at which the $\text{SO}l_0$ solution has a maximum amplitude. We notice a phase drift and small damping as well. For faster results, we increased the time increment to $\tau = 0.5$.

While both solutions $u(x,t)$ are still quite close together at 100 iterations, there is a phase difference that becomes noticeable after 250 iterations and can no longer be denied after 500 iterations. Since we see the $SOl_0$ solutions $u(x,t)$ at their turning point, we can notice that the second-order features not only a phase error but also inherent damping as well. We only realise it apparently later in time evolution in this simple setting. Let us note, however, that the use of non-uniform grids as appearing in higher dimension over a shapes' surface may possibly enhance this phase-shift effect.

5.2.2. Details for the Gaussian Initial Condition

We want to discuss the jigged curves in Figure 4 now. Every time a numerical solver comes across a discontinuity, over- and undershooting is supposed to occur. This is the so-called Gibbs phenomenon. For more details on this topic, we refer the reader to [24].

To illustrate the Gibbs phenomenon, we will solve the $\delta$-peak (cf. Figure 7) and Gaussian initial conditions (cf. Figure 8) with the undamped wave equation and our three numerical methods.

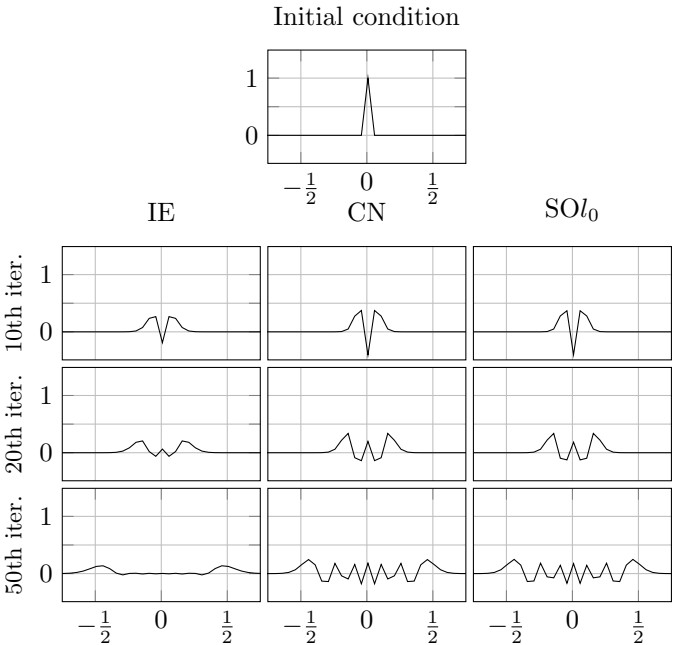

**Figure 7.** (**First row**) We see a delta peak as the initial condition. (**Columns**) The solutions of different numerical schemes (left to right) for the IE, CN and $SOl_0$ schemes are presented. (**Rows 2–4**) We see the solution after different iterations. In the second row, we calculated 10 iterations. The third row shows us the solution after 20 iterations, and in the fourth row, the 50th iteration is presented. All numerical schemes show these oscillations, called the Gibbs phenomenon. This is more visible in the second-order schemes than in the first-order method. The time increment was set to $\tau = 0.01$.

In Figure 7, we computed up to 50 iterations (rows) with our three numerical solvers (columns). For all three methods, we observe two waves moving away from each other. Additionally, we notice over- and undershoots occurring in the centre of the plot, exactly where the discontinuities were. Through the inherent damping of the IE scheme, the Gibbs phenomenon is less distinct and is vanishing faster than in the second-order methods.

In Figure 8, we used a Gaussian curve with $\sigma = \frac{10}{\sqrt{2}}$ as the initial condition. Again, we solved the wave equation with our models. This time we did not observe any over- and undershoots. We only notice the two waves propagating away from each other.

Since we started with a curve with a height of one, both second-order schemes produce two waves with a height of a half. The first-order method shows some loss in the height of the two waves, which indicates again the presence of a numerical damping mechanism.

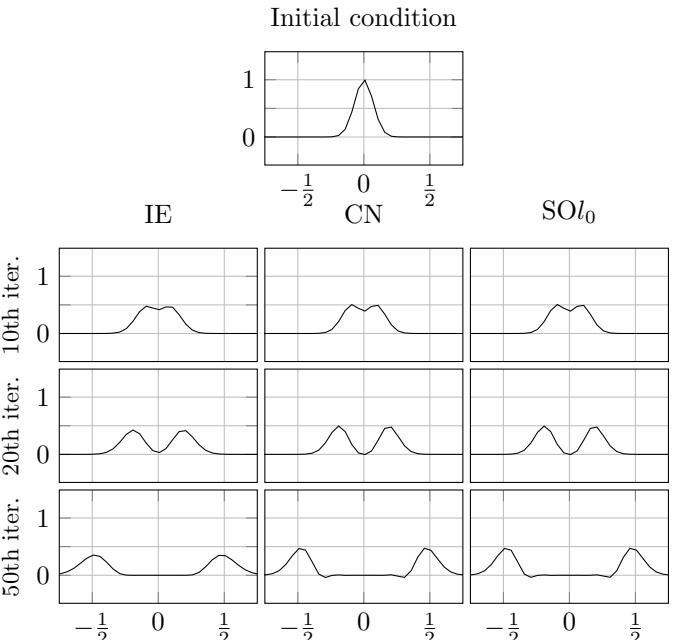

**Figure 8.** (**First row**) We see the Gaussian distribution $\exp\left(-x^2/0.01\right)$ as the initial condition. (**Columns**) The solutions of different numerical schemes for the (left to right) IE, CN and SO$l_0$ schemes are presented. (**Rows 2–4**) We see the solution after different iterations. In the second row, we calculated 10 iterations. The third row shows us the solution after 20 iterations, and in the fourth row, the 50th iteration is presented. The time increment was set to $\tau = 0.01$.

One may conjecture that using the Gaussian initial conditions together with the second-order methods could lead to more information stored in the feature descriptor and therefore to better matching results, as more spatial information might have been collected by the spread initial function compared to the $\delta$ peak.

## 6. Experiments on Shapes

In this part, we aim to transform the ideas and insights of the last section onto real shapes. We will discuss the importance of the damping parameter and Gaussian initial condition as model parameters. For this, we use the settings made in Section 4.3.

Every time we say hit rate, we refer to the hit rate at a geodesic error of 0.25. In all the following plots, we can distinguish the different numerical schemes by colour. The IE method will always be red, CN will be blue, and SO$l_0$ will have the colour green.

### 6.1. Study of the Damping Parameter

First, we wanted to study the impact of the damping parameter. For this experiment, we used 30 different, logarithmic equidistant distributed, damping parameters $k$ between $10^{-3}$ and 1. Then, we computed the hit rate depending on $k$ within a shape class. The classes we chose are cat, centaur, dog, horse and wolf. Additionally, we showed the mean of all classes. To compare the results with our previous works [6,7,9,10], we added $k = 0$ (dashed lines) to simulate the non-damping scenario. The results can be viewed in Figure 9.

Across all classes, we see a similar behaviour of the hit rates. For the first-order method, the hit rate falls with rising $k$. In contrast, the second-order methods increase the hit rate until $k \approx 0.1$. This point differs from shape to shape. For $k > 0.1$, hit rates are the same and fall similar to each other.

Let us take a closer look at the plot from the centaur (cf. Figure 10). In addition, we resized the hit rate axes, so that variations become more evident. Here we notice that the second-order methods have a dent at $k \in (0.01, 0.03)$. While the CN method continues to rise, the SO$l_0$ method decreases in this interval. This inconstant growth in the hit rate can

be observed in all other classes in Figure 9, as well. It is even noticeable in the plot with the mean hit rate.

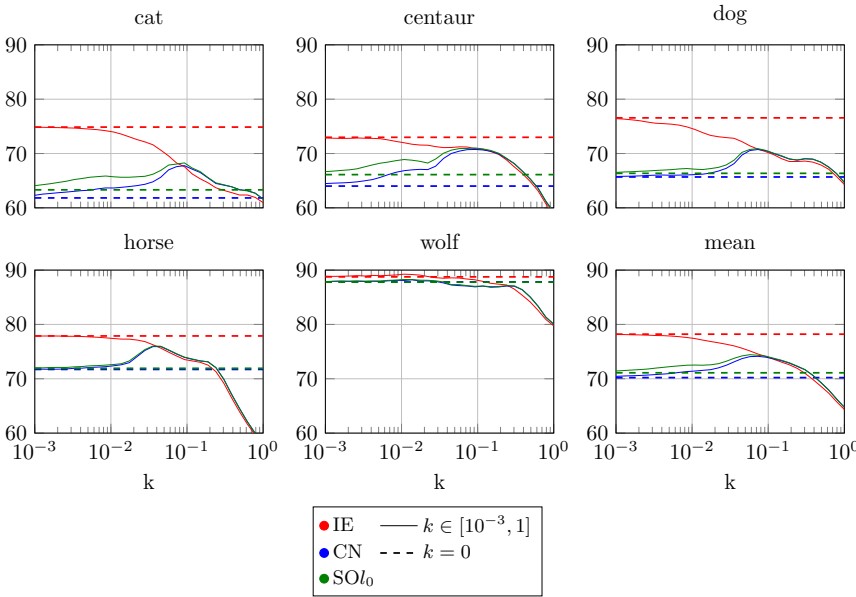

**Figure 9.** We plotted the hit rate at a geodesic error of 0.25 over the damping parameter $k$. With the three different colours, we identify the numerical schemes that were used to solve the damped wave equation on the different shapes. Red indicate the IE scheme, blue denotes the CN method, and green stands for the SO$l_0$ scheme. Solid lines denote a changing damping parameter $k$, and the dashed lines indicate a fixed $k = 0$.

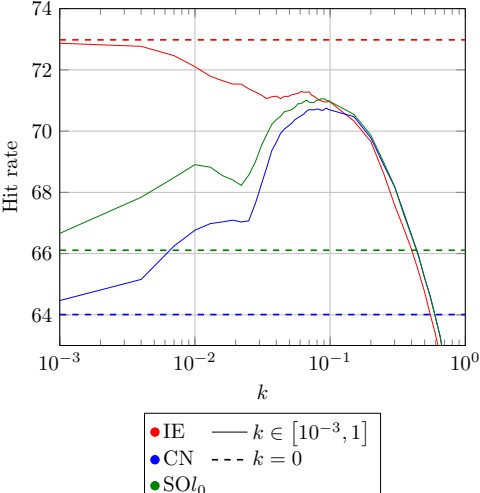

**Figure 10.** Hit rate from the centaur shapes over different damping parameters $k$. Solid lines indicate changing damping parameters and dashed lines denote a fixed damping parameter ($k = 0$). The colours distinguish between the different numerical methods: red for IE, blue for CN and green for SO$l_0$.

In conclusion of these experiments, we see that certain damping is a beneficial mechanism (when about $k \approx 0.1$). Since the second-order methods are supposed to give an accurate account of the underlying analytical model, this means that a reasonable analytical model appears to be the damped wave equation (with $k \approx 0.1$). Let us note in this context that one cannot expect that the results of the IE method and an analytical model resolved at second-order accuracy coincide exactly, so one may interpret in many cases such as in Figure 10 the peak situation (there again at $k \approx 0.1$) and the IE method results as numerically identical.

### 6.2. Gaussian Initial Condition

As we studied in Section 5.2.2, we can expect that the Gaussian initial conditions will provide smoother solutions $u(x, t)$. The reduction of the Gibbs phenomenon should be more beneficial for the second-order methods than for the IE method.

To study this, we recreated the settings from Section 6.1 and changed the initial condition from the $\delta$-peak to the Gaussian distributed peak. We studied three different width parameters: $\sigma = \{\varepsilon, 1, 5\}$. Here, $\varepsilon$ is a small non-negative number and indicates the limit of $\sigma \to 0$. Since our calculations were done in MATLAB, we would like to mention that MATLAB has a built-in machine $\varepsilon$ with a value of $\varepsilon_{\text{machine}} \approx 2.2204 \cdot 10^{-16}$.

As in the previous section, we changed the damping parameter $k$ to be between $10^{-3}$ and 1. The results can be seen in Figure 11. Again, we denote the different numerical solvers with the known colour scheme and keep the meaning of the dashed and solid lines. New to this plot are the symbols on the solid lines, which indicate the different values of $\sigma$. The triangle belongs to $\sigma \to 0$, the square to $\sigma = 1$ and with the circle, we indicate $\sigma = 5$.

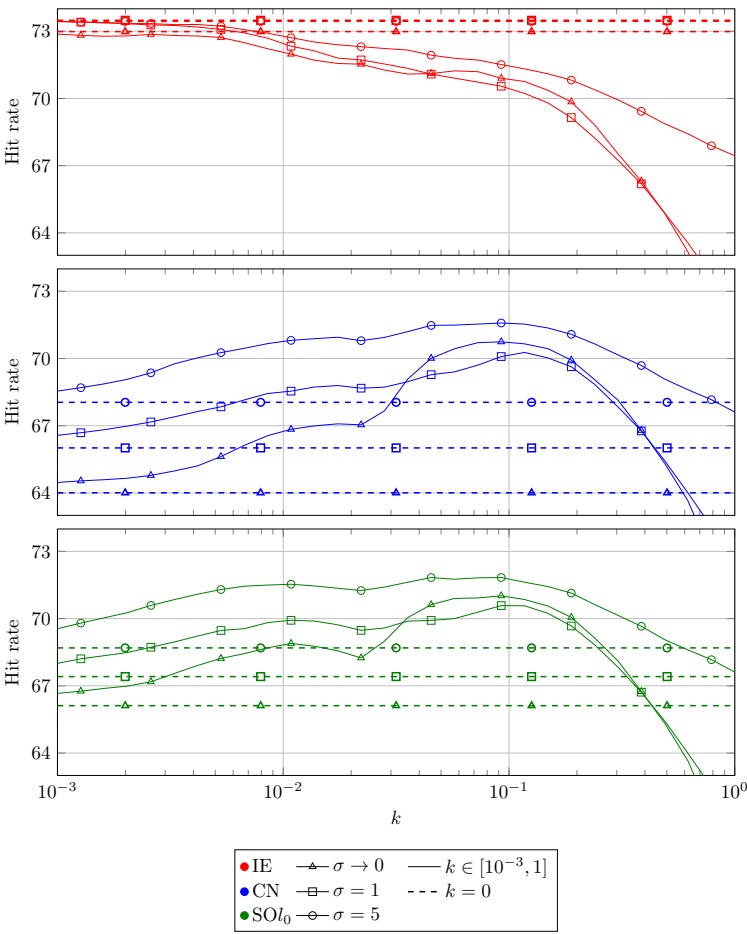

**Figure 11.** (**Top**) to (**bottom**) We see the hit rate at a geodesic error of 0.25 from the centaur shapes obtained by three different numerical schemes. The colours and rows distinguish between the different solvers we used: (red, (**top**)) IE, (blue, (**middle**)) CN and (green, (**bottom**)) SO$l_0$. Solid lines indicate a changing damping parameter $k$ and dashed lines for a fixed ($k = 0$) one. The symbols distinguish between the different widths $\sigma$ of the Gaussian distributed initial condition.

First, we want to mention that the results for $\sigma \to 0$ are similar to the results obtained with the $\delta$-peak initial conditions (cf. Figure 10). According to the theory, this should be the case, but it is a good sign that our calculations verify this.

Second, we notice that in both scenarios, with or without damping ($k = 0$), the hit rate increases with rising $\sigma$. There are two exceptions to this. One is the IE method. A resized version of this plot can be found in Figure 12. Here, the Gaussian initial condition with

$\sigma = 1$ is slightly better than the Gaussian initial condition with $\sigma = 5$. The second exception can be observed in the interval of $k \in [0.03, 0.4]$, where $\sigma \to 0$ performs slightly better than $\sigma = 1$. However, in the end, this means a nearly negligible plus of $\approx 1\%$.

For the second-order schemes, we can still notice the maximum value for the hit rate around $k = 0.1$. Even the changes in $\sigma$ did not affect this value.

In the end, we can conclude that all numerical methods benefit from the Gaussian initial condition. The second-order schemes profit even more than the first-order method.

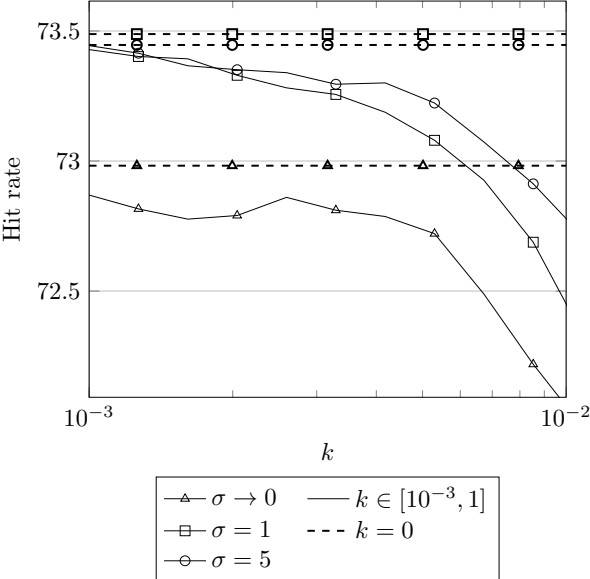

**Figure 12.** Resized version from Figure 11. We only plotted the IE method between $k = 10^{-3}$ and $k = 10^{-2}$. We resized the hit rate axes as well. With these adjustments, it is possible to distinguish the small changes caused by changing the width of the Gaussian initial condition.

### 6.3. Changing Width in Gaussian Initial Conditions

In Section 6.2, we saw that increasing $\sigma$, the width of the Gaussian initial condition, will lead to better hit rates. We wanted to refine these results by fixing the damping parameter to $k = 0.1$ and changing $\sigma$ from 1 to 20. The results can be seen in Figure 13.

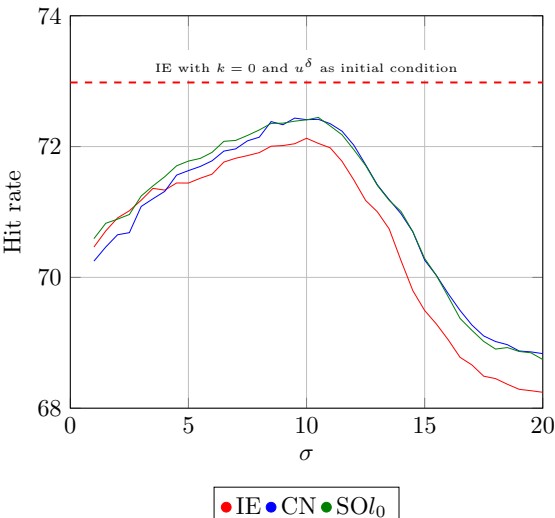

**Figure 13.** The hit rate of the centaur shape over different widths $\sigma$ of the Gaussian initial condition. For this, we fixed the damping parameter at $k = 0.1$. The red solid line denotes the IE method, the blue indicates the CN method and the green line is for the $\text{SO}l_0$ method. The red dashed line shows us the hit rate of the wave equation solved using the IE method and $\delta$-peak initial conditions.

As in the previous plots, we notice an increase in the hit rate. At $\sigma \approx 10$, we reached the maximum value for the hit rate. The maximum value for the IE method is at $\approx 72\%$ and for CN and $SOl_0$ at $\approx 72.4\%$. Again, the difference between the second- and first-order methods is not that large. We added a dashed red line to the plot to indicate the hit rate from the IE method solving the wave equation ($k = 0$). This setting belongs to the best-performing scenarios thus far; here, it reached a hit rate of $\approx 73.25\%$.

As the reported hit rates are partially very small in their differences, one may consider again close solutions as numerically identical.

*6.4. Feature Descriptors with Optimised Parameters*

After we studied the influence of the damping parameter $k$ and width of the Gaussian initial condition $\sigma$ on the hit rate and, therefore, its suitability as a model parameter, we wanted to know their impact on the feature descriptor.

To meet this aim, we recreated the plots from Figure 4, but this time we used the damped wave equation with and without the Gaussian initial condition. In all cases, we used the optimal parameters found during the last sections.

We started with Figure 14. Here, we solved the damped wave equation with a damping parameter of $k = 0.1$ and a $\delta$-peak initial condition. Overall, we notice that the feature descriptors are more smooth compared to the ones in Figure 4. For the second-order methods, we still have some jagged parts at the beginning. These are the remains of the Gibbs phenomenon. In the end, the feature descriptors of the second-order methods are more close to each other, which was not the case in Figure 4. For the hoof, all feature descriptors of the different numerical solvers nearly look alike.

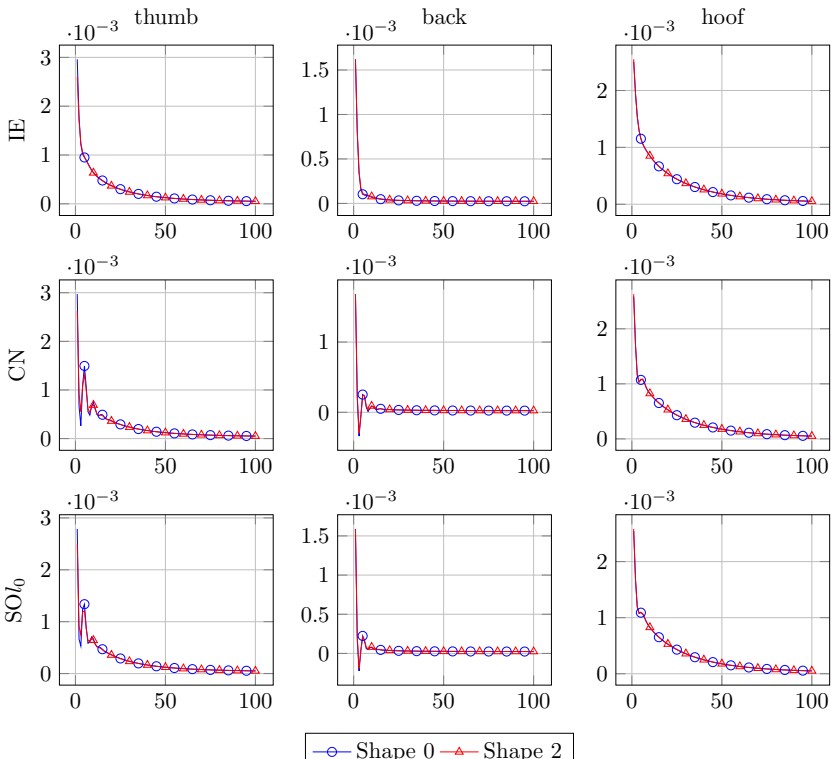

**Figure 14.** The comparison of the feature descriptors from the centaur shape. We used the damped wave equation ($k = 0.1$) with the $\delta$-peak initial condition to generate the feature descriptor. Shape 0, Shape 2 and the points are equal to the points, and shapes shown in Figure 1. (**Rows**) Different solvers. (**Columns**) Different points.

In Figure 15, we solved the damped wave equation with the Gaussian initial condition. The damping parameter was set to $k = 0.1$, and the width of the Gaussian distribution was

set to $\sigma = 10$. The results are similar to those from Figure 14. The main difference is the disappearance of the Gibbs phenomenon remains. For the second-order methods, we still have some artefacts at the back, but there are less than in Figure 14.

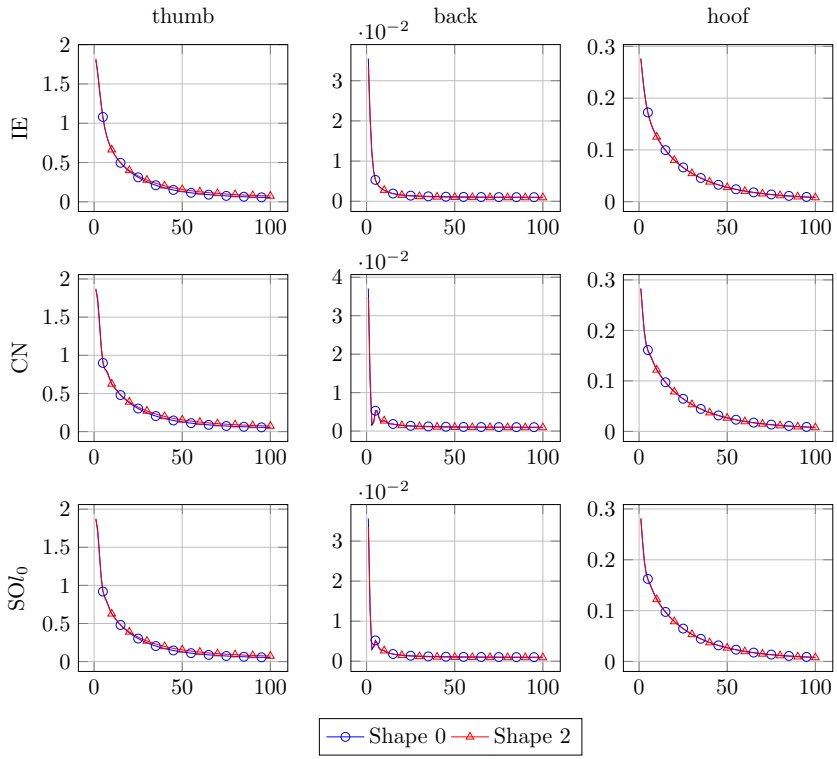

**Figure 15.** The comparison of the feature descriptors from the centaur shape. We used the damped wave equation ($k = 0.1$) with the Gaussian ($\sigma = 10$) initial condition to generate the feature descriptor. Shape 0, Shape 2 and the points are equal to the points, and shapes shown in Figure 1. (**Rows**) Different solvers. (**Columns**) Different points.

In the end, it appears that the optimised feature descriptors have qualitatively much in common with the feature descriptors obtained from solving a heat equation ([10], Figure 4, top left), roughly resembling an exponential damping. This is, however, not too surprising, as we enforce by use of the damped wave equation and Gaussian initial condition important properties such as, e.g., the $l_0$ stability, shared with heat equation results. Let us also note that the obtained hit rates clearly surpass the hit rates one may obtain by use of the pure heat equation or HKS, so it is still evident that wave propagation is an important model ingredient.

### 6.5. Noisy Shape Experiments

The shapes from the TOSCA dataset [2] are artificially created. Therefore, they have no noise in the coordinates of the points. We want to address this fact and add noise to the points. Again, we use the centaur as the shape class for our considerations. We applied a percentage normally distributed noise of 6% and 12% to each shape.

With these noisy shapes, we repeat the experiments with the damping parameter from Section 6.1. This means we solve the damping wave equation with the $\delta$-peak initial condition and repeat this with different damping parameters $k$ between $10^{-3}$ and 1.

The results are plotted in Figure 16. The relations of the colours to the numerical solvers remain the same. Dashed lines still refer to the non-damping scenario ($k = 0$) and the solid lines indicate the change of $k$. However, the symbols on the solid lines refer to the percentage of noise now. The triangle denotes a noise of 6%, and the square denotes a 12% noise.

The first thing we observe is that with higher noise, the hit rate is decreasing, which appears quite natural. The other thing we notice is that, in this scenario, the optimal value for the damping parameter is $k = 0.05$, which is half of the value from the non-noise scenario. What is also somewhat interesting is that we can increase the hit rate for the IE method with certain damping parameters compared to the non-damping case. With a noise of 12% and around $k = 0.02$, we are slightly better compared to the non-damping ($k = 0$) case. The second-order methods show similar behaviour as in Figure 10.

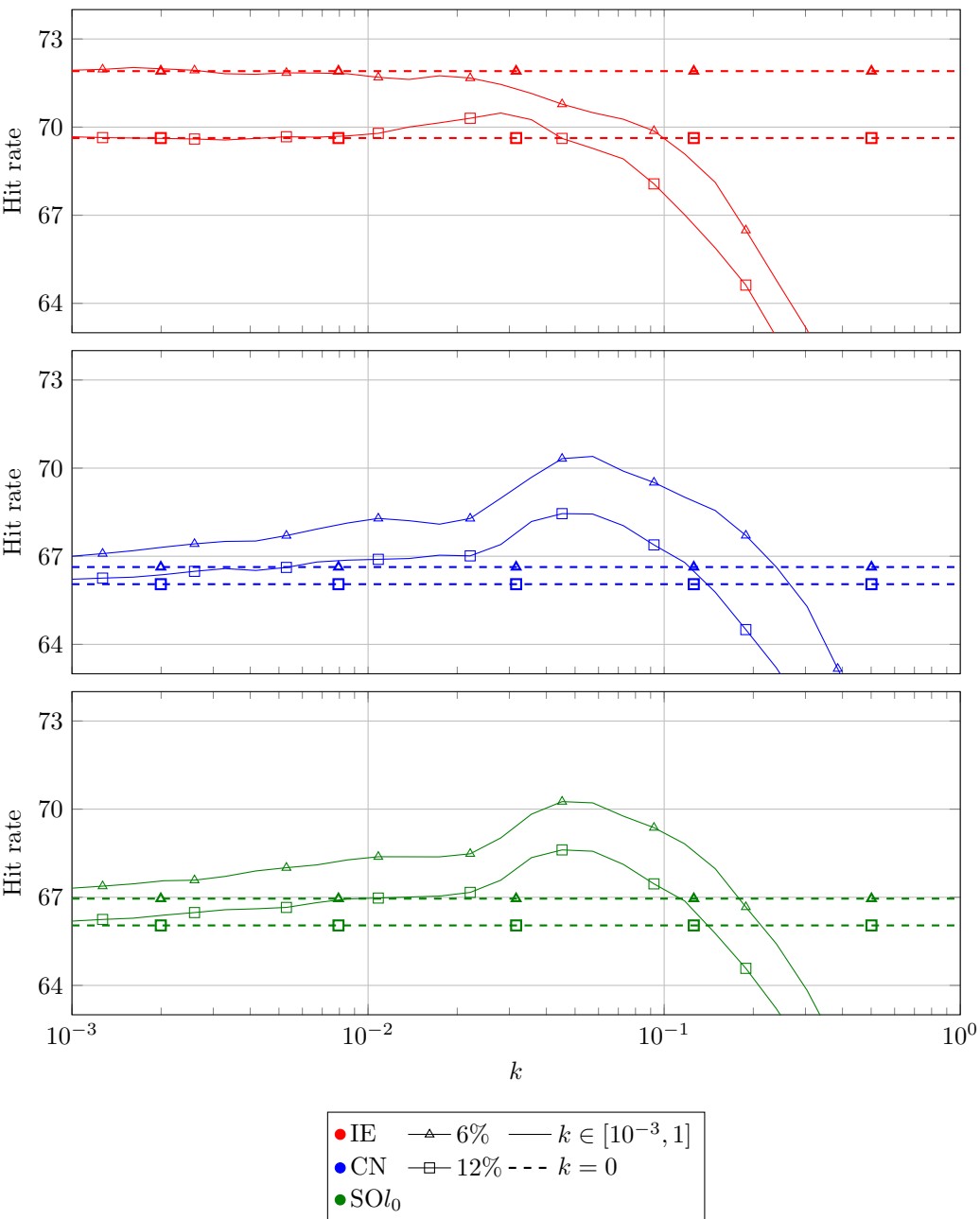

**Figure 16.** We solved the damped wave equation using different numerical schemes and the $\delta$-peaked initial condition over different $k$. Then, we computed the hit rate at a geodesic error of 0.25 and plotted the hit rate over different damping parameters $k$. From (**Top**) to (**Bottom**), we see the hit rate produced with the IE method (red), the CN scheme (blue) and the $SOl_0$ method (green). With the dashed line, we indicate the results from the $k = 0$ scenario. The triangle and square denote a noise of 6% and a noise of 12%, respectively.

Next, we wanted to recreate the experiments from Section 6.3. This time, we fixed the damping parameter at $k = 0.05$ and solved the damped wave equation with the Gaussian distributed initial condition. We repeated this with different values for $\sigma$, the width of the Gaussian distribution. As before, we change $\sigma$ to be between 1 and 20.

The results are illustrated in Figure 17. The meaning of the colours and the symbols did not change. We obtained similar results for both noise levels. In the beginning, we see an increase in the hit rate. When entering the interval $\sigma \approx 7$ to $\sigma \approx 12$, we notice a plateau in the hit rate. In this interval, a rising $\sigma$ did not affect the hit rate. For $\sigma \approx 12$, the hit rate is decreasing again.

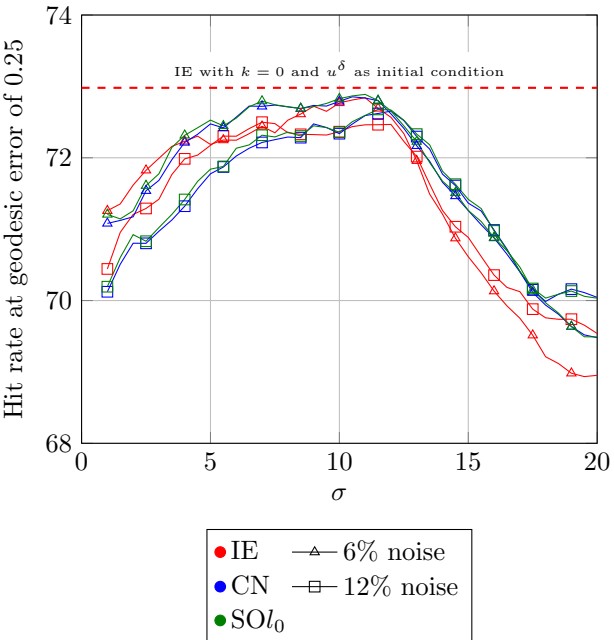

**Figure 17.** We solved the damped wave equation ($k = 5 \cdot 10^{-2}$) with the Gaussian initial condition. Then, we computed the hit rate at a geodesic error of 0.25 and plotted it over different values of $\sigma$. Additionally, we plotted the hit rate from the wave equation without damping obtained with $\delta$-peak initial conditions and solved by the IE method as a red dashed line. The triangle and square denote a noise of 6% and a noise of 12%, respectively.

Overall, our results from the noisy shapes are similar to the results from the artificial shapes. This indicates that our approach is robust and could be transferred to real-world shape-matching scenarios.

## 7. Conclusion and Further Work

This paper represents an attempt to study the influence of several discrete and continuous-scale model properties for the shape-matching scenario. In particular, we analysed in detail that a (certain, but not too large) damping mechanism is a useful model property, which may relate to the analogon of $l_0$ stability of the underlying analytical model.

The usage of the $\delta$-peak condition was motivated by the works on the HKS [4] and WKS [5]. In our approach to the shape-matching framework, the Gaussian distributed initial conditions seem favourable. In all experiments, we see an increase in the hit rate when using Gaussian distributed initial conditions.

One of the most interesting points for future work seems to us to put a focus on the influence of the initial condition, since it appears that it is important for hit rate quality what kind of initial signal will be propagated. The experiments reported here using the Gaussian initialisation may represent a starting point for this.

**Author Contributions:** Conceptualization, A.K. (Alexander Köhler) and M.B. (Michael Breuß); methodology, A.K. and M.B.; software, A.K.; validation, A.K. and M.B.; formal analysis, A.K.; investigation, A.K.; resources, M.B.; data curation, A.K.; writing—original draft preparation, A.K.; writing—review and editing, A.K. and M.B.; visualization, A.K.; supervision, A.K. and M.B.; project administration, A.K. and M.B. All authors have read and agreed to the published version of the manuscript.

**Funding:** This research received no external funding.

**Institutional Review Board Statement:** Not applicable.

**Informed Consent Statement:** Not applicable.

**Data Availability Statement:** The data presented in this study are openly available from [2].

**Conflicts of Interest:** The authors declare no conflict of interest.

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
