# Peer review of "Computational Analysis of PDE-Based Shape Analysis Models by Exploring the Damped Wave Equation"

_algorithms, doi:10.3390/a15090304_

Round 1

Reviewer 1 Report

Some suggestions to the authors:

Let me to suggest that in order to analyze shapes in different positions, i.e. regardless of their embedding in the 3D space, we could recur to the Egregium Theorem of Gauss. Actually, what remains invariant in a 2D or 3D shape, regardless of its location and position in space, is the Gaussian curvature, that could be evaluted with numerical schemes. Moreover, a pointwise analysis can make the results dependent on the chosen points-

Furthermore, for closed forms, the Fourier descriptors could also be used, which would have the advantage of regardless by the number of selected points and the points themselves.

Giuliani Donatella

Author Response

We have included references to incorporate the above suggestions.

Reviewer 2 Report

The problem of shape correspondences computation is studied in this work. Here the problem is solved by using numerical solution of partial differential equations on the manifold which shape is under consideration. Authors choose the damped wave equation  and justify this choice. The authors use three numerical solution methods that are applied to the Сentaur shape and other shapes of humans and animals. A comparative analysis of the obtained numerical results is carried out in order to choose the optimal method for solving the problem. 

 The problem is very important in computer vision. The work is interesting, the arguments are mostly presented in detail, the conclusions are justified. The following few remarks relate to the introductory mathematical background of the work.  

 1. What are $\alpha_{ij}$, $\beta_{ij}$ in (18)?

 2. The line after (23). What properties of $W$ will lead to such excellent properties of eigenvalues and eigenvectors of problem (23)? 

 3. How did the second equation in (24) turn out? 

 4. In the right-hand side of (29) the index $G$ is lower. But in (22) it was the upper index. 

 5. What is kind of publication [8]?

Author Response

  1. We have added a description of alpha and beta. Furthermore, we have referred to the right-hand image of Figure 3.
  2. That is a good point. In fact, we were not detailed enough here. We have added that the property in mention is the symmetry of W. 
  3. That is actually not obvious enough. We have briefly explained the basic steps needed to obtain equations 2 and 3 in (24). 
  4. Thank you for pointing this out. This typo has been corrected.
  5. The reference [8] was indeed missing some information. We have updated this. Thank you for the information.